# Transcriptomic Analysis and Functional Gene Expression in Different Stages of Gonadal Development of *Macrobrachium rosenbergii*

Zhenxiao Zhong [1], Guozhu Chen [1], Haihui Tu [1], Xinyi Yao [1], Xin Peng [1], Xuan Lan [1], Qiongying Tang [1,*], Shaokui Yi [1], Zhenglong Xia [2], Miaoying Cai [2] and Guoliang Yang [1,2,*]

1 Key Laboratory of Aquatic Animal Genetic Breeding and Nutrition, Chinese Academy of Fishery Sciences, Zhejiang Provincial Key Laboratory of Aquatic Resources Conservation and Development, Huzhou University, Huzhou 313000, China
2 Jiangsu Shufeng Prawn Breeding Co., Ltd., Gaoyou 225654, China
* Correspondence: tangqy@zjhu.edu.cn (Q.T.); ygl0572@163.com (G.Y.)

**Abstract:** In order to decipher the functional genes and reveal the molecular mechanism of gonadal development in *Macrobrachium rosenbergii*, a comparative transcriptome analysis was performed on the testes and ovaries at different developmental stages. A total of 146,537 unigenes with an N50 of 2008 bp and an average length of 1144 bp were obtained from the sequencing raw data via quality control and denovo assembly. Identification of differentially expressed genes (DEGs) showed that there were 339 and 468 DEGs among the different developmental stages of testes and ovaries, respectively, and 7993 DEGs between the testes and ovaries. The KEGG enrichment analysis identified 13 candidate pathways related to gonadal development, including insulin synthesis, oocyte maturation, and steroid biosynthesis, which were involved in biological processes such as regulation of hormone metabolism, sex cell proliferation and development, and amino acid metabolism. The DEGs related to the above pathways such as *insulin-like growth factor 1 receptor* (*IGF1R*), *heat shock protein 90* (*Hsp 90*), and *cyclooxygenase* (*COX*) genes were highly expressed during yolk protein synthesis, indicating that these genes might be involved in yolk accumulation and oogenesis. Meanwhile, *calmodulin* (*CaM*) and other genes were highly expressed during spermatogenesis, suggesting that these genes might play an important role in spermatogenesis. Ten differentially expressed genes in the KEGG signaling pathway, including *CRQ*, *COX*, *APP*, *Cdc42*, *Hsd17b12*, *Art-1*, *Hsp70*, *Hsp90*, *PRMT1*, and *GP*, were selected for real-time quantitative PCR (RT-qPCR) to validate the transcriptome data, and the results showed that RT-qPCR obtained consistent results with the RNA-Seq data. The present findings provide new insights into the molecular regulation mechanism of gonadal development in *M. rosenbergii*.

**Keywords:** *Macrobrachium rosenbergii*; gonadal development; transcriptome; differentially expressed genes; signaling pathway



## 1. Introduction

The giant freshwater prawn (GFP) *Macrobrachium rosenbergii*, belonging to the phylum Arthropoda, order Decapoda, family Palaemonidae, and genus *Macrobrachium*, is native to tropical and subtropical regions, and is the largest cultured freshwater shrimp in the world. Because of its high economic value, short culture cycle, wide-ranging diet and other advantages, it has become an important economic species of freshwater shrimps. China introduced this species for farming in 1976 and has become its largest producer in the world, as China's GFP production accounts for 50–60% of the world's total production [1]. As an introduced species in China, there are some limitations such as a small effective population size, difficulty in obtaining post-larvae, and an unstable supply of high-quality

post-larvae, resulting in an imbalance between the supply and demand of good post-larvae for GFP farming [2,3].

Growth and reproductive performance, as two major economic traits, can directly affect the efficiency and progress of the selective breeding of targeted varieties, thus affecting the development of the industry. Zhao [4] and Wang [5] investigated the structures of reproductive systems in *M. rosenbergii* females and males, respectively, and preliminarily staged the process of oogenesis and spermatogenesis; Zhu et al. [6] analyzed the expression of the *foxl*2 gene in the hepatopancreas and ovaries before and after sexual maturation in *M. rosenbergii*. These studies have provided us with a preliminary understanding of the structure of the reproductive system, gonadal development process and related functional genes in *M. rosenbergii*, but the molecular mechanisms of gonadal development and regulatory mechanisms still remain unknown. Currently, artificial nursing is often characterized by a lack of synchronization of female spawning, leading to an uneven larval catch, a longer nursing cycle, and higher costs [7]. Further investigations of the basic rules of gonadal development for *M. rosenbergii* can help to improve the synchronization of spawning by artificial intervention at key points of growth and reproduction, thus helping to improve the size and fertility of spawning individuals, and enhance the survival rate of post-larvae, which is an important prerequisite for culturing good breeding stocks.

In recent years, transcriptome sequencing has been widely used for the study of gene expression patterns in crustaceans [8], especially for non-model organisms that lack genomic information. In gonad-related transcriptomic studies, many authors have investigated the molecular mechanisms of gonadal development in cultured shrimps and crabs through transcriptome sequencing or by using candidate genes related to sex differentiation and gonadal development, and they also have explored more unknown related functional genes via transcriptome sequencing. For example, Wei et al. [9] identified genes related to gonadal development and maturation through a large-scale transcriptome study of the gonads in *Litopenaeus vannamei*, Jiang et al. [10] annotated a total of 36,282 genes related to sex in *M. rosenbergii* through gonadal transcriptome analysis, and detected *gametocyte-specific factors* (*Gtsf*), *heat shock protein 21* (*HSP21*), *male reproduction-related peptidase inhibitor Kazal-type* (*MRPINK*) and 23 potential sex-related candidate genes, in addition to several sex-related KEGG pathways that also found. In *Macrobrachium nipponense* [11] and *Eriocheir sinensis* [12], researchers compared transcriptomic data between normally mature and precocious individuals to elucidate the phenomenon of precocious maturation, and screened for genes related to precocious maturation such as *Vitellogenin* (*Vg*), *cyclooxygenase* (*COX*), and *glutathione peroxidase* (*gpx*) in *M. nipponense*. In addition, some authors performed a comparative transcriptome analysis of gonads at different developmental stages. For example, Chen et al. [13] obtained 4592 differentially expressed genes via transcriptome analysis of two key developmental stages in *Lysmata vittate* and found some genes including *insulin-like androgenic hormone* (IAG), *crustacean female sex hormone* (CFSH), and *gonad inhibiting hormone* (GIH) as well as some pathways such as Wnt and ovarian steroid hormone biosynthesis that may be involved in their sex differentiation and gonadal development. Waiho et al. [14] identified the most significantly differentially expressed gene *β lens protein-like* via transcriptome analysis of the testes at three developmental stages for *Scylla olivacea*. Transcriptomic analysis of different developmental stages of gonads has become an important way to study the molecular and regulatory mechanisms of crustacean development.

Based on external morphology observations and a conventional histologic section of the *M. rosenbergii* gonads at different development stages, we have previously divided the development of its ovaries into eight periods and its testes into four periods [15]. In order to further explore the functional genes related to gonadal development and understand their expression patterns, we selected several key periods of gonadal development with obvious differences, and performed transcriptome analysis on the ovaries and testes. The present study will provide a theoretical basis for the selection of breeding stocks and new insights

into the molecular regulation mechanism of gonadal development, further enriching the reproductive biology of *M. rosenbergii*.

## 2. Materials and Methods

### 2.1. Experimental Materials

Prawns used in the present study were the same age and were selected from nine families from the Breeding Center of Jiangsu Shufeng Prawn Breeding Co., Ltd. (Gaoyou, China). A total of 200 individuals at an age of around 80 days were selected to be cultured from each family, and the ratio of male to female was controlled at about 1:1. In total, 1800 prawns were cultured for investigating the external morphology and internal histology of the gonads (our published paper [15]) and for utilization in the transcriptome analysis in this study. Prawns from the nine families were placed separately in nine concrete ponds with an area of about 17 $m^2$, and all culture conditions including the light (0~1380 lx), water temperature (28 °C~32 °C), and feed were kept consistent across all ponds. Water quality parameters were maintained at pH 7.5~8, dissolved oxygen 8~11 mg/L, ammonia nitrogen 0.05~0.25 mg/L, and nitrite 0.1~0.3 mg/L. Sampling was performed every week from 1 August 2020 to 4 October 2020, and each pond was sampled. Three males and three females were randomly caught for each sample from each pond, and the gonads were observed and sampled.

According to the division and identification of the gonadal developmental stages of *M. rosenbergii* as defined by Chen et al. [15], three individuals each with ovaries at Stage I, III, VI, VII, and VIII, and testes at Stage I, II, and III, were randomly sampled from the nine families and preserved in RNA-later for total RNA extraction and transcriptome sequencing.

### 2.2. Total RNA Extraction and Transcriptome Sequencing

Total RNA was extracted from each prawn's gonad by using the conventional Trizol method. An ND-1000 spectrophotometer (NanoDrop, Wilmington, DE, USA) and 1% agarose gel electrophoresis were used to detect the RNA quality and concentration. The concentration of total RNA should be greater than 250 ng/μL with an OD260/OD280 value between 1.8 and 2.2 and an OD260/OD230 value greater than or equal to 2.0 to ensure no degradation and contamination of the RNA. The qualified total RNA was used for the following library construction and transcriptome sequencing, which was performed by Beijing Novogene Bioinformatics Technology Co. LTD. The sequencing platform Illumina Hiseq 2500 (Illumina, San Diego, CA, USA) was used.

### 2.3. Sequence Assembly and Gene Function Annotation

To ensure the quality and reliability of the data analysis, the raw sequencing data were controlled for quality by removing adapter-containing reads, reads including poly-N, and low-quality reads to obtain reliable, clean reads. Trimmomatic v0.35 (USADELLAB, USA) [16] was used for raw read trimming. Three independent software modules, cocoon, pupa and butterfly, which are based on Trinity-v2.4.0 software (The Broad Institute, Boston, MA, USA) [17], were adopted to splice the filtered clean reads to obtain transcripts (-min_kmer_cov: 3). On this basis, the redundancy was removed through Corset-version 4.6 (Nadia M Davidson, Alicia Oshlack, 2014) hierarchical clustering. Then, the length of transcripts and cluster sequences were counted, and the transcript sequence was used as the reference sequence. The longest transcript in each gene was used as the unigene. The BUSCO-v5.4.4 software (Benchmarking Universal Single-Copy Orthologs) (SIB, Geneva, Switzerland) [18] uses a single-copy orthologue gene library to assess the completeness of the assembled transcripts in combination with tblastn [19], augustus [20], and hmmer software [21]. To obtain more comprehensive gene function information, the unigenes were compared and annotated through seven public databases: NCBI Nt (non-redundant nucleotide sequences), NCBI Nr (non-redundant protein sequences), Pfam (protein families),

KOG (EuKaryotic Orthologous Groups), Swiss-prot, KEGG (Kyoto Encyclopedia of Genes and Genomes), and GO (Gene Ontology) [22].

### 2.4. Identification of Differentially Expressed Genes, and GO and KEGG Enrichment Analysis

In this study, the FPKM (fragments per kilobase of exon model per million mapped fragments) method was used to calculate the expression of genes [23]. Differential expression analysis of two groups was performed using the DESeq2 R package (1.20.0). DESeq2 provides statistical routines for determining the differential expression in digital gene expression data using a model based on the negative binomial distribution. The criteria for screening differentially expressed genes (DEGs) are $|\log_2 (\text{fold change})| > 1$ and FDR (false discovery rate) $< 0.05$ [24]. Under the threshold of a $p$-value $\leq 0.05$, GO [25] and KEGG [26], enrichment analyses of DEGs were performed using GOSeq (1.10.0) (GAP TECH, Beijing, China) and KOBAS (v2.0.12) (Center for Bioinformatics, Peking University, China) packages, respectively. Moreover, their annotation information from the concomitant KEGG pathways, GO, and Pfam library databases was analyzed for a conclusive identification. Heatmaps visualizing gene differential expression were created using Gene Denovo.

### 2.5. Real-Time Quantitative PCR Validation of Differentially Expressed Genes

Based on the results of the transcriptome analysis, 10 DEGs were selected for real-time quantitative PCR (RT-qPCR) validation. The specific primers (Table 1) for each gene were designed using the Primer 6.0 software (Premier Biosoft, Palo Alto, CA, USA) and synthesized by Wuhan Tianyi Huayu Gene Technology Co., Ltd. The reverse transcription reaction system was a total of 30 μL, including 6 μL of 5 X PrimeScript RT Master Mix, 15 μL of total RNA, and 9 μL of RNase Free ddH$_2$O. The reaction program included 37 °C for 15 min, then 85 °C for 5 s, and the obtained cDNA was stored at 4 °C for the following RT-qPCR, which was performed using the SYBR PreMix Ex Taq kit (TaKaRa, Dalian, China) with 18S rRNA as the reference gene. Three technical replicates were set up for each sample, and the reaction system totaled 25 μL, including 12.5 μL of TB Green Premix Ex *Taq* II (2×), 9.5 μL of ddH2O, 1 μL each of forward and reverse primers, and 1 μL of cDNA. The CFX96 Touch$^{TM}$ RT-qPCR instrument (Bio-Rad, Hercules, CA, USA) was used. Based on the $2^{-\Delta\Delta CT}$ calculation method, the relative expression of the target gene in the samples was obtained, and the significance test was carried out using SPSS 25.0 (IBM, Chicago, IL, USA), and compared with the transcriptome sequencing data.

**Table 1.** Primers used for the RT-qPCR analysis of the gonadal development in *Macrobrachium rosenbergii*.

| Number | Gene Name | Sequence 5′-3′ | Length (bp) |
|:---:|:---:|:---:|:---:|
| 1 | *CRQ* | F: CCGCAGAACGCAAAGAGAAT | 228 |
| | *Croquet protein* | R: TGACAGTTGGACAGCAGACA | 214 |
| 2 | *COX* | F: TCATCCGTGGCAGTTCTTGT | 251 |
| | *Cyclooxygenase* | R: GCACTTGTGGCATCTGGTATC | 224 |
| 3 | *APP* | F: GATGATGACGACGACGATGAC | 201 |
| | *Amyloid-beta-like protein* | R: TCAGGTGGAGTAGAGGCATTG | 283 |
| 4 | *Cdc42* | F: CAGACCATCAAGTGCGTAGTG | 256 |
| | *Cell division control protein 42* | R: CACGGAGAAGCAGACCAGAA | 201 |
| 5 | *Hsd17b12* | F: CCTACACTTATGGCTCCAACAC | 249 |
| | *Hydroxysteroid (17-beta) dehydrogenase 12a* | R: CTCCCTCTCCTTCCTCTTCAAA | 234 |
| 6 | *Art-1* | F: CGTTGCTTACCATGTGAACCA | 262 |
| | *CoA reductase-1* | R: TCCAAGAACCAACCTCGTATGT | 249 |
| 7 | *Hsp70* | F: CCAAGCAGACTCAGACATTCAC | 216 |
| | *Heat shock protein 70* | R: CAGCAGACACATTCAGGATACC | 282 |
| 8 | *Hsp90* | F: AACTGTCTCGCTCCACTTGA | 228 |
| | *Heat shock protein 90* | R: TACCTTCCTCCTCATCTTCCTC | 239 |
| 9 | *PRMT1* | F: TTCTCCTCTACGCCGCTTT | 200 |
| | *Protein arginine methyltransferases 1* | R: GTTCCGCAACCAACATCCA | 224 |

**Table 1.** *Cont.*

| Number | Gene Name | Sequence 5′-3′ | Length (bp) |
|---|---|---|---|
| 10 | *GP* | F: GCTCAACGACACTCATCCTTC | 207 |
| | *Glycogen phosphorylase* | R: CGCCTGCTAACTTCCTGTAGA | 203 |
| 11 | 18S rRNA | F: TATACGCTAGTGGAGCTGGAA | 286 |
| | | R: GGGGAGGTAGTGACGAAAAT | 257 |

## 3. Results

### 3.1. Transcript Assembly and Statistical Analysis

Transcriptome sequencing was performed on 15 ovarian samples from five different developmental stages of females, and a total of 352,975,807 raw reads were obtained, with an average of 23,531,720 raw reads per sample. A total of nine testicular samples from three different developmental stages of males were sequenced, and 210,202,884 raw reads were obtained, with an average of 23,355,876 raw reads per sample. After filtering, a total of 328,723,887 and 201,496,918 clean reads were obtained for the ovaries and testes, respectively, with a corresponding average of 21,914,926 and 22,388,546 per sample. Together, the clean data of each sample were more than 5.90 Gb, and all values of the Q20 and Q30 percentages were higher than 98% and 94%, respectively. The proportion of clean reads accounted for more than 94% of the raw data, and the N50 was 2008 bp (Table 2). The filtered data were assembled with Trinity software, and a total of 146,537 unigenes were obtained after hierarchical clustering with Corset, with a sequence length ranging from 301 bp to 28,406 bp and an average length of 1144 bp. A total of 60,907 unigenes had a length of 300–500 bp, accounting for 41.56% of the total number of unigenes. The second largest number of unigenes was 1001–2000 bp in length, numbering 20,749 and accounting for 14.16% of the total number of unigenes; 19,879 unigenes had a length of more than 2000 bp, accounting for 13.57% of the total number of unigenes (Figure 1). The BUSCO evaluation showed that the transcript completeness was 91%, indicating that the quality of data assembly was good.

**Table 2.** Quality of transcriptome sequencing and assembly of the gonads in *Macrobrachium rosenbergii*.

| Sample ID | Tissue | Raw Reads | Clean Reads | Clean Bases | Error (%) | Q20 (%) | Q30 (%) | GC Content (%) |
|---|---|---|---|---|---|---|---|---|
| AC1 | STI ovary | 22,801,902 | 22,362,483 | 6.71G | 0.02 | 98.1 | 94.37 | 41.95 |
| AC2 | STI ovary | 23,113,581 | 22,616,983 | 6.79G | 0.02 | 98.18 | 94.43 | 41.41 |
| AC3 | STI ovary | 20,423,838 | 20,034,059 | 6.01G | 0.03 | 98.06 | 94.04 | 41.75 |
| DC1 | STIII ovary | 23,366,938 | 22,945,237 | 6.88G | 0.02 | 98.22 | 94.4 | 41.51 |
| DC2 | STIII ovary | 22,622,979 | 22,155,554 | 6.65G | 0.02 | 98.4 | 94.89 | 41.64 |
| DC3 | STIII ovary | 20,350,586 | 19,954,445 | 5.99G | 0.02 | 98.25 | 94.48 | 41.34 |
| FC1 | STVI ovary | 22,058,708 | 21,518,653 | 6.46G | 0.03 | 98.06 | 94.11 | 41.23 |
| FC2 | STVI ovary | 21,638,453 | 21,203,487 | 6.36G | 0.02 | 98.17 | 94.29 | 41.17 |
| FC3 | STVI ovary | 23,790,503 | 23,267,866 | 6.98G | 0.02 | 98.38 | 94.78 | 41.51 |
| GC1 | STVII ovary | 21,968,333 | 21,489,585 | 6.45G | 0.02 | 98.32 | 94.64 | 40.99 |
| GC2 | STVII ovary | 24,898,145 | 24,301,643 | 7.29G | 0.02 | 98.34 | 94.73 | 41.98 |
| GC3 | STVII ovary | 22,953,131 | 22,439,268 | 6.73G | 0.02 | 98.17 | 94.34 | 41.84 |
| ZC1 | STVIII ovary | 20,212,619 | 19,796,150 | 5.94G | 0.02 | 98.29 | 94.58 | 41.68 |
| ZC2 | STVIII ovary | 22,706,643 | 22,141,590 | 6.64G | 0.02 | 98.3 | 94.71 | 41.93 |
| ZC3 | STVIII ovary | 23,083,488 | 22,496,884 | 6.75G | 0.02 | 98.24 | 94.57 | 41.68 |
| AX1 | STI testis | 23,730,986 | 23,280,534 | 6.98G | 0.02 | 98.25 | 94.54 | 40.55 |
| AX2 | STI testis | 22,838,934 | 22,222,520 | 6.67G | 0.02 | 98.23 | 94.52 | 41.11 |
| AX3 | STI testis | 23,593,669 | 23,074,458 | 6.92G | 0.02 | 98.37 | 94.81 | 40.24 |
| BX1 | STII testis | 23,037,584 | 22,343,012 | 6.70G | 0.02 | 98.27 | 94.61 | 42.25 |
| BX2 | STII testis | 22,223,635 | 21,787,130 | 6.54G | 0.02 | 98.21 | 94.34 | 39.27 |
| BX3 | STII testis | 22,571,163 | 22,255,874 | 6.68G | 0.02 | 98.12 | 94.06 | 41.27 |
| DX1 | STIII testis | 23,243,749 | 22,870,217 | 6.86G | 0.02 | 98.18 | 94.26 | 39.77 |
| DX2 | STIII testis | 22,337,164 | 21,865,987 | 6.56G | 0.02 | 98.24 | 94.54 | 41.8 |
| DX3 | STIII testis | 22,273,017 | 21,797,186 | 6.54G | 0.02 | 98.19 | 94.44 | 42.53 |

Note: STI, STII, STIII, STVI, STVII, and STVIII represent gonadal development Stage I, II, III, VI, VII, and VIII, respectively. Reference for staging of ovaries and testes: Chen et al. [14].

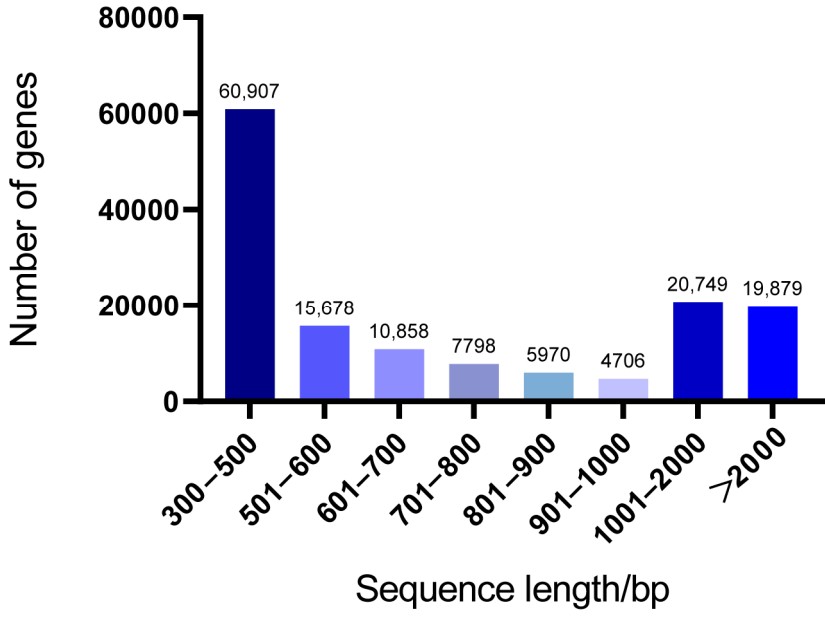

**Figure 1.** Distribution of unigene length of the transcriptome in the gonads of *Macrobrachium rosenbergii*. The darker the color, the larger the number.

### 3.2. Functional Annotation and Classification of the Unigenes

The 146,537 unigenes obtained were annotated by using a homology search against seven public databases, including Nr, Nt, KO, Swiss-prot, Pfam, GO, and KOG, and 27,684 (18.89%), 7474 (5.10%), 8781(5.99%), 15,032 (10.25%), 30,530 (20.83%), 30,520 (20.82%), and 7278 (4.96%) unigenes were annotated in the corresponding database (Table 3). The 2148 genes are co-annotated by the five major databases, including Nt, Nr, KOG, GO and Pfam. Aligning the unigene sequences of *M. rosenbergii* with the Nr database, it was found that the sequence similarity with *Litopenaeus vannamei* was the highest, reaching 52.9%, and had a similarity of only 3.6%, 3.5%, and 2.0% with *Mizuhopecten yessoensis*, *Hyalella azteca*, and *Apostichopus japonicus*, respectively. The sequence similarity between *M. rosenbergii* and *Armadillidium vulgare* was the lowest at only 1.5%, indicating that the unigenes of the gonad transcriptomes in *M. rosenbergii* has a high homology with *L. vannamei*.

**Table 3.** Statistics of unigene annotations based on each of the seven databases.

| Database for Annotation | Number of Unigenes | Percentage of Annotated Unigenes (%) |
|---|---|---|
| Annotated in Nr | 27,684 | 18.89 |
| Annotated in Nt | 7474 | 5.10 |
| Annotated in KO | 8781 | 5.99 |
| Annotated in SwissProt | 15,032 | 10.25 |
| Annotated in Pfam | 30,530 | 20.83 |
| Annotated in GO | 30,520 | 20.82 |
| Annotated in KOG | 7278 | 4.96 |
| Annotated in all databases | 2148 | 1.46 |
| Annotated in at least one database | 43,427 | 29.63 |
| Total unigenes | 146,537 | 100 |

Based on the annotated results from the GO database, 30,520 unigenes were classified into 42 subcategories of three GO categories including "Biological processes", "Cellular components", and "Molecular functions". "Biological process" was the largest category, in which two subcategories, "Cellular processes" and "Metabolic process", contained the largest number of unigenes at 17,872 and 14,495, respectively. Out of the "Cellular component" category, the subcategory "Cellular anatomical entity" consisted of the largest

number of unigenes (15,176). In the "Molecular function" category, 14,449 unigenes were in the "Binding" subcategory, and 10,548 unigenes were related to "Catalytic activity" subcategory (Figure 2).

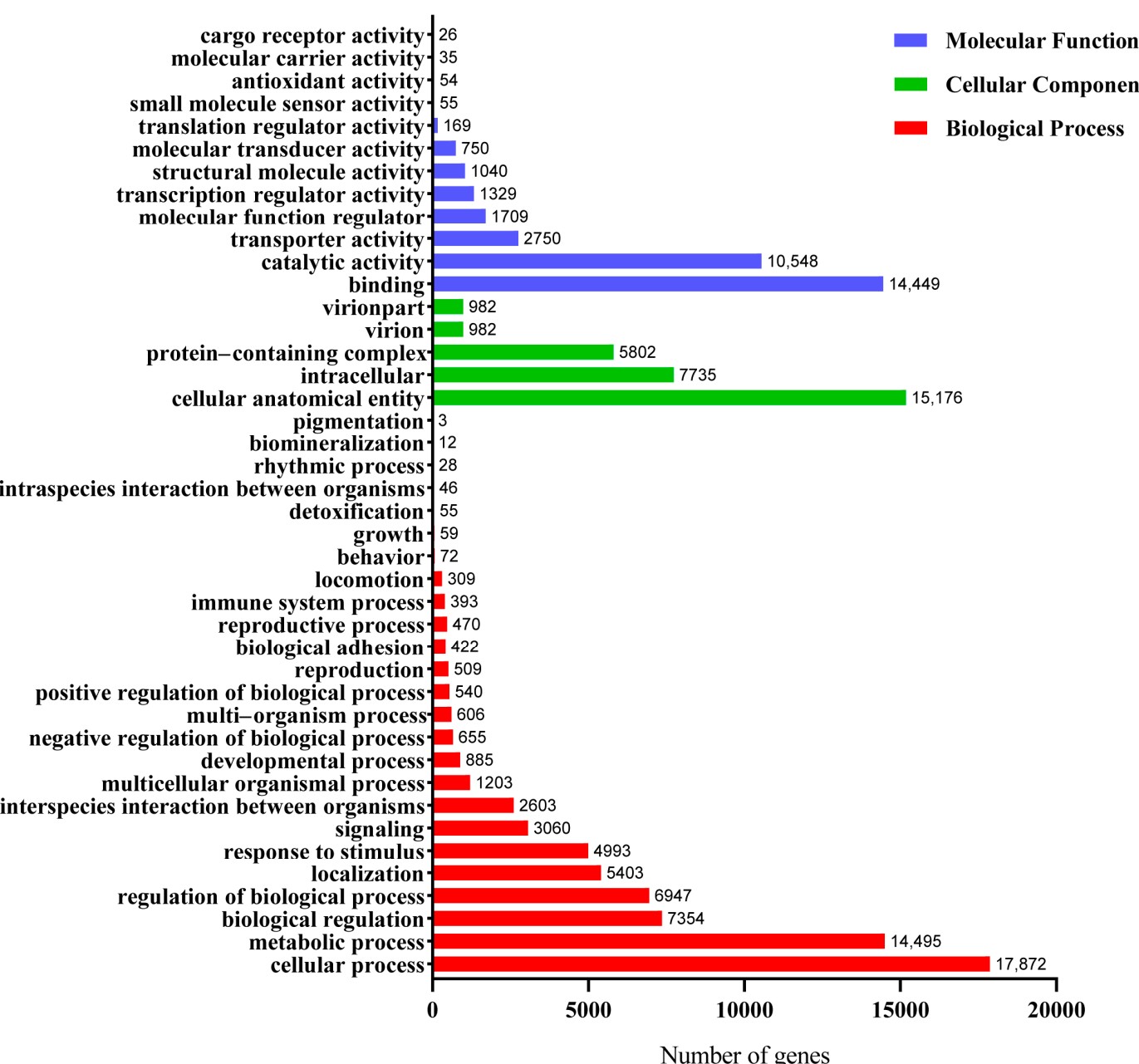

**Figure 2.** GO annotation of the gonad transcriptome of *Macrobrachium rosenbergii*.

The KEGG database annotation results showed that a total of 8901 unigenes were classified into 35 different secondary pathways of 5 branches (branch A to E), among which the top three secondary pathways were "Signal transduction", "Transportation and catabolism" and "Endocrine system", where the number of unigenes included were 1084, 724, and 574, respectively (Figure 3).

### 3.3. Identification of Differentially Expressed Genes

The analysis of DEGs in the ovaries of *M. rosenbergii* at different developmental stages showed that the number of DEGs gradually increased with the development of the ovaries. There were 11,775 DEGs between the Stage I and Stage III ovaries (AC vs. DC), and

12,434 DEGs between the Stage I and Stage VII ovaries (AC vs. GC). However, the number of DEGs decreased to 10,184 when the ovaries finished ovipositing at the degenerative stage (Stage VIII), as compared to Stage I (AC vs. ZC). The data from the male testes showed an increasing number of DEGs as the testes developed, from 7277 between Stage I and Stage II (AX vs. BX) to 9197 between Stage I and Stage III (AX vs. DX) (Figure 4).

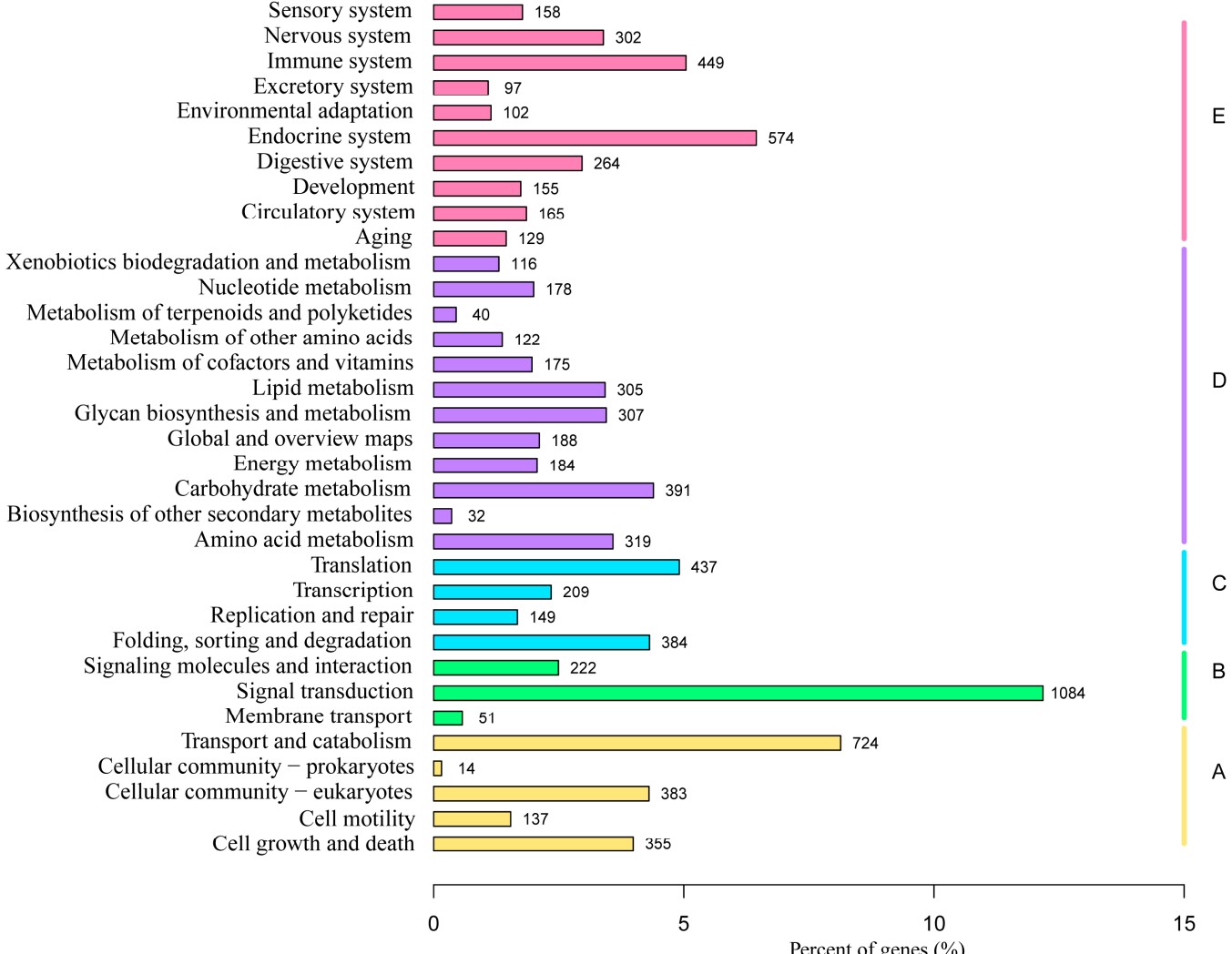

**Figure 3.** Unigene KEGG enrichment classification of the gonad transcriptome of *Macrobrachium rosenbergii*.

The number of DEGs between the testes and ovaries were much more than that between ovaries or testes at different development stages, and also increased as the ovaries and testes developed, changing from 20,879 between Stage I ovaries and testes (AC vs. AX), to 22,338 at Stage II (DC vs. BX), and finally, to 28,593 at Stage III (GC vs. DX) (Figure 4).

Venn diagrams including the up- and down-regulated genes of three pairwise comparison showed that there was a total number of 339 common DEGs in the testes of male *M. rosenbergii* at different developmental stages (Figure 5a). There were 468 common DEGs among the top 5 DEG (the number of DEGs > 5000) pairwise comparison combinations between different development stages of female ovaries (Figure 5b). Moreover, a total of 7993 common DEGs were identified to be among the top 4 DEG (the number of DEGs > 20,000) pairwise comparison groups between testes and ovaries (Figure 5c).

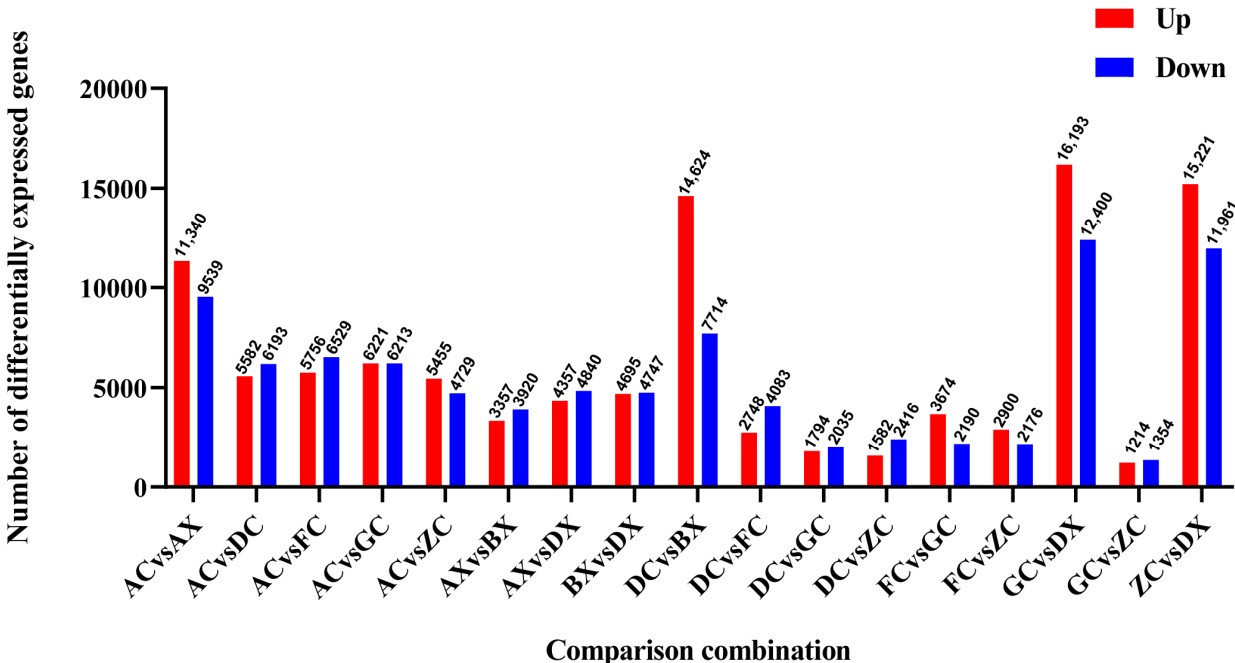

**Figure 4.** Statistics of differentially expressed genes in different stages of gonadal development of *Macrobrachium rosenbergii*. AC—Stage I ovary; DC— Stage III ovary; FC—Stage VI ovary; GC—Stage VIII ovary; ZC—Stage VIII ovary; AX—Stage I testis; BX—Stage II testis; DX—Stage III testis.

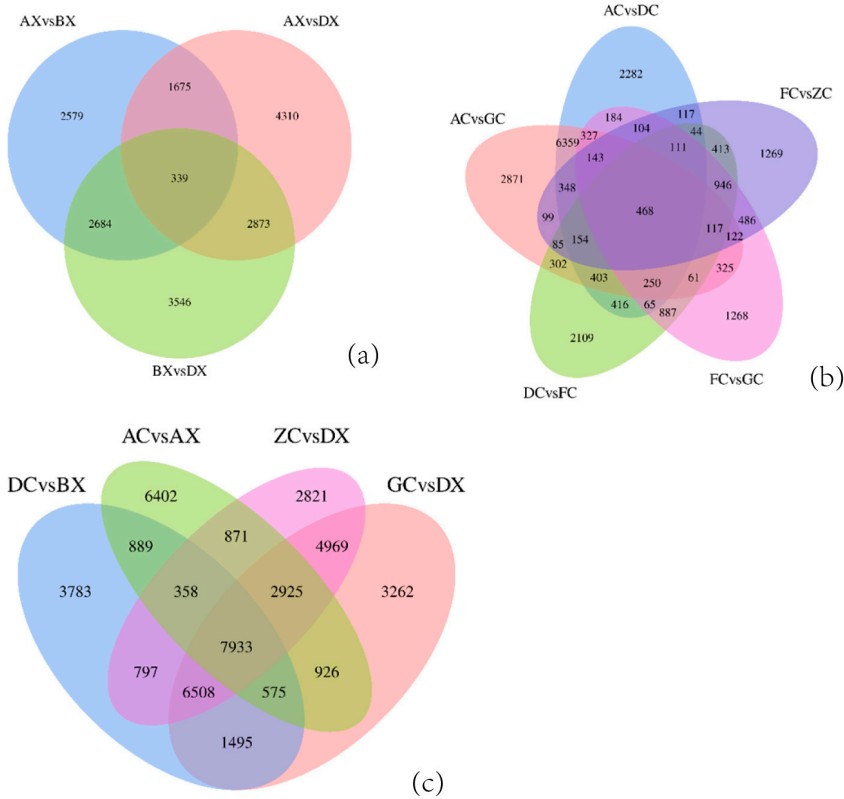

**Figure 5.** Venn diagram of differentially expressed genes in different gonadal development stages of *Macrobrachium rosenbergii*. (**a**): three pairwise comparisons of the testes; (**b**): the top 5 DEG pairwise comparisons of ovaries; (**c**) the top 4 DEG pairwise comparisons between testes and ovaries. AC—Stage I ovary; DC—Stage III ovary; FC—Stage VI ovary; GC—Stage VII ovary; ZC—Stage VIII ovary; AX—Stage I testis; BX—Stage II testis; DX—Stage III testis.

*3.4. KEGG Enrichment Analysis*

Based on the KEGG enrichment analysis of the DEGs shared by testes or ovaries at different development stages, we found 13 KEGG signaling pathways that might be related to testis or ovary development; the pathways consisted of steroid biosynthesis, ovarian steroidogenesis, pantothenic acid and CoA biosynthesis, insulin secretion, fatty acid biosynthesis, the GnRH signaling pathway, steroid hormone biosynthesis, progesterone-mediated oocyte maturation, oocyte meiosis, unsaturated fatty acid biosynthesis, the insulin signaling pathway, the estrogen signaling pathway, arginine biosynthesis, and other signaling pathways. Among them, the insulin signaling pathway (involving 69 overlapping DEGs), oocyte meiosis (including 65 overlapping DEGs), the estrogen signaling pathway (59 common DEGs), and progesterone-mediated oocyte maturation (58 overlapping DEGs) were the top four pathways with the highest number of shared DEGs (Figure 6).

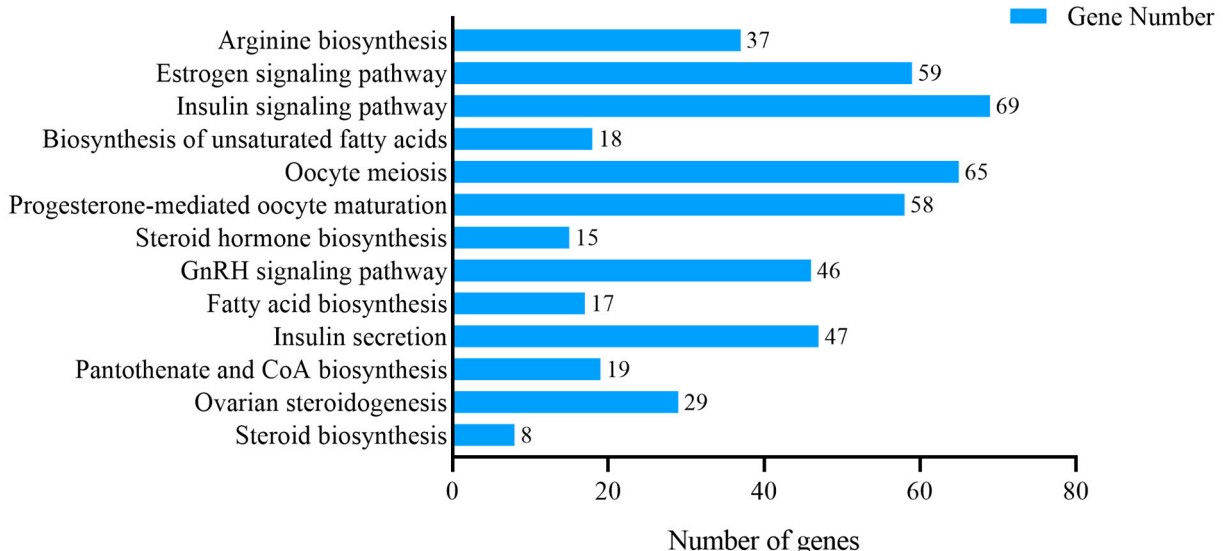

**Figure 6.** The 13 KEGG pathways that might be related to the regulation of gonadal development in *Macrobrachium rosenbergii*. The value next to each bar is the number of the enriched common DEGs.

To investigate the relationship between different development stages and DEGs, 20 DEGs related to gonadal development were selected for DEG clustering analysis to show the expression pattern of DEGs. Heat maps of DEGs demonstrated that most genes were differentially expressed between testes and ovaries, and especially between ovaries in Stages III (DC), VI (FC), VII (GC), and VIII (ZC) and testes in each developmental period (AX, BX, and DX) (Figure 7). For example, the *calmodulin* gene (*CaM*) was consistently highly expressed in the developing testes of male, but was much less expressed in the developing ovaries. The *croquet protein* (*CRQ*), *glycogen phosphorylase* (*GP*), and *insulin-like growth factor I receptor* gene (*IGF1R*) were consistently highly expressed in the ovaries from the endogenous yolk synthesis stage (DC) to the maturation (ZC) of the ovaries, but was lowly expressed in the testes. The *cell division regulatory protein 42* gene (*Cdc 42*) was highly expressed at the beginning of ovary development in female *M. rosenbergii*, whereas it was lowly expressed at the beginning of testis development. In addition, some genes were differentially expressed at different stages of testis or ovary development. For example, the *cyclooxygenase* gene (*COX*) and *amyloid-beta-like protein* (*APP*) were highly expressed when the female ovary enters the rapid developmental stage (FC), and the *protein arginine methylesterase* gene (*PRMT1*) was highly expressed when the ovary matured (GC), but they were lowly expressed during other periods of ovary development. For *heat shock protein 90* (*Hsp 90*) gene members, *Hsp 90-2, Hsp 90-4, and Hsp 90-5* were consistantly highly expressed after the initiation of yolk protein synthesis (DC) in the ovaries of *M. rosenbergii*, but was lowly expressed in the testes. Figure 8 shows the detailed expression level of several common EDGs in the KEGG signaling pathway related to gonadal development.

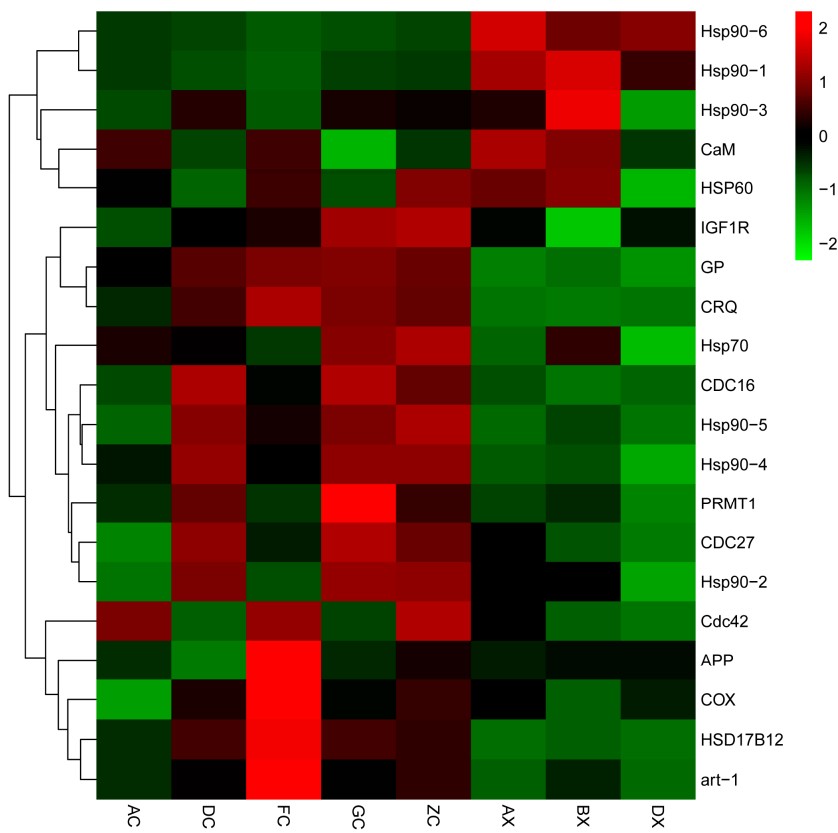

**Figure 7.** Clustering and gene module heatmap of common differentially expressed genes in KEGG signaling pathway related to gonadal development. AC—Stage I ovary; DC—Stage III ovary; FC—Stage VI ovary; GC—Stage VII ovary; ZC—Stage VIII ovary; AX—Stage I testis; BX—Stage II testis; DX—Stage III testis.

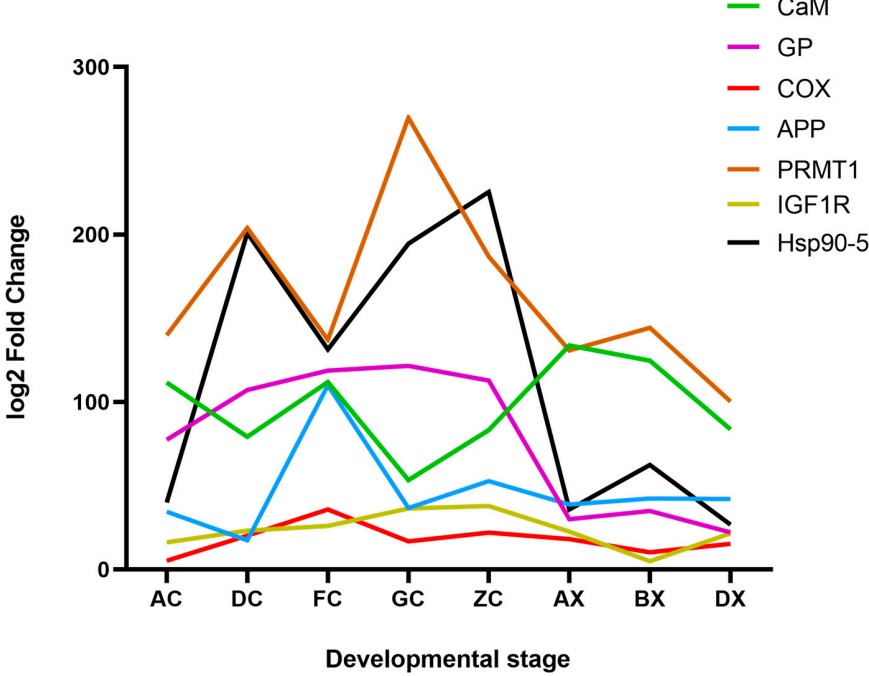

**Figure 8.** Expression level of several common differentially expressed genes in KEGG signal pathway related to gonadal development. AC—Stage I ovary; DC—Stage III ovary; FC—Stage VI ovary; GC—Stage VII ovary; ZC—Stage VIII ovary; AX—Stage I testis; BX—Stage II testis; DX—Stage III testis.

*3.5. RT-qPCR Validation*

Ten DEGs in the KEGG signaling pathway associated with gonadal development were selected for RT-qPCR validation. The selected genes included *croquet protein* (*CRQ*), *cyclooxygenase* (*COX*), *amyloid-beta-like protein* (*APP*), *cell division control protein* 42 (*Cdc 42*), *hydroxysteroid (17-beta) dehydrogenase 12a* (*Hsd17b12*), *enoyl-CoA reductase-1* (*Art-1*), *heat shock protein 70* (*Hsp 70*), *heat shock protein 90* (*Hsp 90*), *protein arginine methyltransferases 1* (*PRMT 1*), and *glycogen phosphorylase* (*GP*). RT-qPCR data are shown in Figure 9. The five down-regulated genes in the RNA-seq, *CRQ, COX, Cdc 42, Art-1* and *Hsp 90*, were down-regulated in RT-qPCR as well, and the five up-regulated genes in the RNA-seq, *APP, Hsd17b12, Hsp 70, PRMT 1,* and *GP*, were up-regulated in the RT-qPCR validation. The validation results were basically consistent with the expression levels of the corresponding genes in the transcriptome, indicating that the transcriptomic data were reliable.

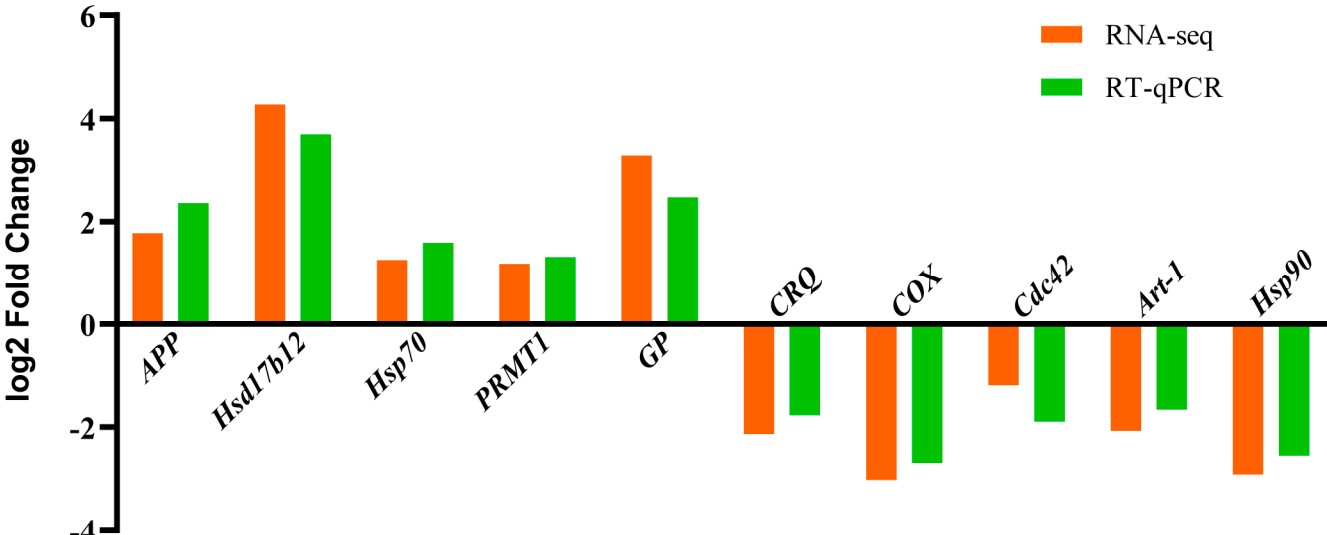

**Figure 9.** Validation of DEGs by RT-qPCR.

## 4. Discussion

### 4.1. Analysis of Gonadal Development-Related Pathways

The KEGG enrichment analysis of metabolic pathways identified 13 signaling pathways related to gonadal development, including insulin synthesis, oocyte maturation, and steroid biosynthesis, which involve biological processes such as the regulation of hormone metabolism, sex cell proliferation and development, and amino acid metabolism. Arginine as an essential component of sperm nucleoprotein is involved in sperm formation in male individuals [27]; in *Salmo trutta*, the secretion level of insulin-like growth factor I (IGF-I) can be raised via an intraperitoneal injection of arginine [28]. Insulin-like growth factors (IGFs) are involved in the regulation of gonadal developmental processes and meanwhile may affect the growth and development of individuals [29,30]. The present transcriptome analysis showed that there were differentially expressed genes in the arginine biosynthesis pathway at different development stages for female ovaries and male testes. Thus, it is hypothesized that arginine biosynthesis might indirectly regulate testis and ovary development by promoting growth and gonadal development-related hormone secretion.

Dynamic changes in animal energy metabolism directly influence the secretion level of reproduction-related hormones, thus affecting individual reproductive performance. Insulin is a key hormone involved in the regulation of energy metabolism, whereas gonadotropin-releasing hormone (GnRH) is an important hormone related to gonadal development and reproduction. Insulin generally regulates gonadal development and reproduction via its impact on the secretion level of GnRH [31], and the regulatory pathways include the following three: (1) promoting the binding of early growth response protein-1 (Egr-1) to the GnRH promoter, in order to up-regulate GnRH promoter activity and the

consequent high GnRH expression [32]; (2) influencing GnRH expression by affecting extracellular signal-regulated kinase (ERK) activity in the presence of follicle stimulating hormone (FSH) and luteinizing hormone (LH) [33,34]; and (3) regulating GnRH expression by enhancing neuropeptide Y (NPY) activity [35]. The present results show that DEGs were detected in insulin- and GnRH-related signaling pathways in testes and ovaries at different developmental stages, indicating that the gonadal development process in *M. rosenbergii* is closely related to the individual growth and energy metabolism of the organism.

Pantothenic acid, switching to coenzyme A (CoA), becomes involved in fatty acid metabolism. CoA is involved in the metabolism of glucose and lipids, providing 90% of the energy for body consumption. In addition, pantothenic acid is also involved in the synthesis and metabolism of steroids, which can be used as substance for fatty acid synthesis. Sex steroid hormones such as progesterone, 17β-estradiol and testosterone are key regulatory hormones of gonadal development. Ovarian steroids were found to have effects on the formation of the yolk, as well as the proliferation, development, and maturation of the zygote in *M. rosenbergii* [36]. Testosterone was found to induce spermatozoal development in male Chinese mitten crab (*Eriocheir sinensis*), whereas 17β-estradiol can induce the transfer of liver nutrients into ovaries in female crabs, accelerating ovarian nutrient accumulation and promoting ovarian development [37]. The results of this study showed that a significantly differential expression existed between the gonads of *M. rosenbergii* males and females in the pathways related to the synthesis of steroids, fatty acids, pantothenic acid and CoA, indicating that these pathways might play an important regulatory role in gametogenesis and yolk accumulation in *M. rosenbergii*.

### 4.2. Key Differentially Expressed Genes Related to Gonadal Development

The present study showed that several genes such as *IGF1R*, *CaM*, *Hsp90*, *COX*, and *PRMT1* might play important roles in regulating gonadal development in *M. rosenbergii*.

IGF1R is an insulin-like growth factor 1 membrane receptor. The insulin-like protein family includes insulin-like peptides (ILPs) and insulin-like growth factors (IGFs) [38,39]. Both IGF-1 and IGF1R are members of the insulin-like protein family. IGF-1 regulates cell mitosis and cell differentiation, participating in growth, sexual differentiation, and the reproduction process [40]. IGF1R acts as a receptor for IGF-I in vertebrates, which can accelerate the proliferation of ovarian mesenchymal cells while promoting DNA synthesis [41]. In invertebrates, such as *Macrobrachium nipponense*, the high-expression of the *IGF1R* gene implies the massive production of IGF-1, which promotes yolk synthesis, oogenesis, and sexual maturation in females [11]. In Drosophila, insulin-like growth factor signaling pathways, including IGF-1 and IGF1R, have been proved to be linked to yolk formation, oogenesis, and juvenile hormone-related reproductive delay [42]. The present results show that the *IGF1R* gene was consistently highly expressed in female *M. rosenbergii* ovaries after entering the phase of endogenous yolk protein synthesis, whereas it was lowly-expressed in ovaries at the early stages, suggesting that IGF1R might contribute to yolk formation in female *M. rosenbergii*.

CaM is a calmodulin, which is involved in cell proliferation and differentiation by binding to Ca$^{2+}$. Studies in amphibian and fish oocytes during early development have shown that CaM is involved in the release of gonadotropin-like hormone in the GnRH signaling pathway by activating the calmodulin kinase (CaM kinase). CaM accelerates oocyte development by promoting the synthesis and expression of factors related to the sex hormone synthesis pathways and oocyte maturation pathways, such as estradiol and progesterone [43]. Studies in mammals have concluded that CaM is closely related to mammalian spermatogenesis, flagellar motility, the acrosome reaction, and sperm–egg fusion [44]. The current results show that the *CaM* gene was highly expressed during the development of testes, but the expression level was low during ovary development, indicating that *CaM* might contribute to accelerating spermatogonia proliferation and maturation. However, its function of inducing egg maturation in female *M. rosenbergii* needs to be further proved.

In oviparous animals, estrogen can upregulate transcriptional levels of the *vitellogenin* (*Vg*) gene and accelerate the accumulation of yolk protein in the ovaries [45]. Researchers have considered that there may be a heat shock protein 90-vitellogenin (HSP90-Vg) molecular regulatory channel, which means Hsp90 can bind to estrogen receptors in the absence of estrogen and perform an estrogen-like function, consequently upregulating the transcription of the *Vg* gene and accelerating the synthesis of Vg [46]. This study shows that some *Hsp90* gene members was consistently highly expressed after the initiation of yolk protein synthesis (Stage III) in the ovaries of *M. rosenbergii*. Therefore, it was hypothesized that the some *Hsp90* gene members might have a significant impact on the accumulation of the yolk protein in female *M. rosenbergii*.

Protein arginine methylesterase (PRMT1), which is responsible for the methylation of protein arginine substrates and protein methylation modifications, is involved in cellular signaling transcription, RNA splicing, transcriptional regulation, and other biological processes [47]. In *Artemia sinica*, the *PRMT1* gene was expressed at all stages throughout embryonic development, peaking at the flourishing period of development. The expression of *PRMT1* gradually decreased in adulthood, suggesting that *PRMT1* can regulate the process of embryonic development [48]. In transcriptomic data from the testes and ovaries of *Oratosquilla oratoria* [49], researchers screened for the *PRMT1* gene as a candidate for germ cell differentiation and development regulation. The present results showed that *PRMT1* gene expression was significantly up-regulated during the rapid developmental stages (Stages IV, V and VI) and significantly down-regulated during the degenerative stage (Stage VIII) in the ovaries of *M. rosenbergii*, suggesting that *PRMT1* might play a vital role in inducing the maturation of ovaries.

Cyclooxygenase (COX) can catalyze the production of prostaglandins (PGs) from arachidonic acid (AA). PGs have effects on ovarian maturation in females [50]. Studies on *Cancer pagurus* have shown that PG levels gradually increases as the ovaries mature [51]. There was a high level of PGs in the early development stages of ovaries in *Panulirus japonicus* and PG levels gradually decreased as the ovaries matured [52]. There was a lower expression of the *COX* gene in premature female *M. rosenbergii* compared to mature individuals, indirectly suggesting that the PG level affects the development of ovaries in *M. rosenbergii* females [11]. This study found that the expression level of the *COX* gene was significantly up-regulated when the ovaries entered the rapid development stage (Stage IV, V, and VI), whereas it was significantly down-regulated in the early stages of ovarian development (Stage I, II, and III) and in the mature stage (Stage VII). The expression of the *COX* gene in the testes always remained low, suggesting that the *COX* gene might be attributed to the acceleration of ovarian maturation in *M. rosenbergii*.

In addition, the present study indicated that *amyloid-beta-like protein* (*APP*) and the *glycogen phosphorylase* (*GP*) also might play an important regulatory role in the gonadal development of *M. rosenbergii*, but few studies have been performed on their functions.

Ultimately, the development of gonads is a complicated process that involves a large number of genes and their regulation. The detailed function of the genes related to gonadal development found in this study need be further proved in future investigations.

## 5. Conclusions

In conclusion, the transcriptomic data obtained from several different developmental periods of ovaries and testes of *M. rosenbergii*, involved 146,537 unigenes, enriching the database of gonadal development-related genes in *M. rosenbergii*. Transcriptomic comparisons of testes and ovaries at different developmental stages helped identify 13 key pathways (steroid biosynthesis, arginine biosynthesis, GnRH signaling pathway, etc.) that might be involved in the gonadal development of *M. rosenbergii*. These pathways are mainly involved in hormone regulation, gonadal cell proliferation, and amino acid metabolism. Importantly, some key genes (*IGF1R, CaM, Hsp90, COX, PRMT1*, etc.) were detected, which might contribute to gonadal developmental regulation, hormone metabolism, gametogenesis, yolk accumulation, etc. The present results can provide basic information for

further investigations on the mechanisms of sex differences and the molecular regulation of gonadal development in *M. rosenbergii*.

**Author Contributions:** Conceptualization, Z.Z. and G.C.; methodology, H.T.; software, X.Y.; validation, X.P. and X.L.; formal analysis, S.Y.; investigation, Z.X.; resources, M.C.; project administration, Q.T.; funding acquisition, G.Y. All authors have read and agreed to the published version of the manuscript.

**Funding:** This research was funded by National Key R&D Programme of China for Blue Granary (number: 2018YFD0901300), The Major Research & Development Programme (Modern Agriculture) of Jiangsu Province (number: BE2019352), The Earmarked Fund (number: CARS-48), Special Funds for Major Science and Technology of Breeding New Agriculture (aquatic) Varieties in Zhejiang province (number: 2021C02069–4-3).

**Institutional Review Board Statement:** This study was conducted according to the Guide for Laboratory Animals developed by the Ministry of Science and Technology (Beijing, China). The experimental protocol for animal care and tissue collection was approved by Huzhou University (the ethical approval code: 20190625).

**Informed Consent Statement:** Not applicable.

**Data Availability Statement:** All supporting data are included within the main article.

**Acknowledgments:** We thank Houkuan Du, Jie Yang, Minmin Yang, and Qianqian Xing for their help in sample and data collection. Thanks are also given to two anonymous reviewers and editors for their valuable advice on revising our manuscript.

**Conflicts of Interest:** The authors declare no conflict of interest.

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
