# Peer review of "Transcriptomic Analysis and Functional Gene Expression in Different Stages of Gonadal Development of Macrobrachium rosenbergii"

_fishes, doi:10.3390/fishes8020094_

Round 1
Reviewer 1 Report
This manuscript presents RNA.seq analyses that aim to decipher the functional genes and reveal the molecular mechanism of gonadal development in the shrimp Macrobrachium rosenbergii. To this end, the authors use testes and ovaries at different developmental stages to extant RNA for the analyses. Among other interesting data, they identify 13 candidate pathways related to gonadal development, including insulin synthesis, oocyte maturation, and steroid biosynthesis, which are involved in biological processes such as regulation of hormone metabolism, sex cell proliferation and development, and amino acid
metabolism. Ten differentially expressed genes in the KEGG signaling pathway were selected for real-time quantitative PCR (RT- qPCR) validations and the results were consistent with the RNA-Seq data. The authors need to detail more the proceedings used for the RNA-seq analyses (e.g. It was not clear if they used R or RStudio). Some images can also be improved such as Fig2 and Fig4, (e.g. colors are very different from one figure to another, not a very consistent graphism throughout the paper). Apart from that, the manuscript is interesting, relevant, and adapted to the readership of the journal.
Author Response
Please see the document attached.

Reviewer 2 Report
The authors sampled several developmentally important stages of gonad development in Macrobrachium rosenbergii (freshwater prawn) and characterized the transcriptome from these samples. They did this in the hopes that understanding gonad development better will lead to improvements in farming the freshwater prawn, and more specifically in fertility and survival of progeny. The authors had previously characterized the developmental stages histologically, and in this study made the next logical step of characterizing the transcription of genes in these tissues. While this study is an important step towards the suggested improvements in fertility and progeny survival, the study as it is has several major limitations.
The major issues that I observed that should be addressed before publication are:
1) The authors present a lot of information, but it is not in a very meaningful way that gives the reader a better understanding of the biology of gonad development in the freshwater prawn. There are many ways to address this issue. I will suggest one, but there are other ways. One way might be to present the information on a developmental time-line, clustering genes based on expression level for each stage (with enriched GO terms for each cluster), and then identifying the differences between each timelines clusters. This way, we get a better understanding of what is unique between each stage, and how it changes through time. Preferably, this would be on one figure for each gonad type.
2) The explanations in the discussion do not seem to be supported by the evidence. The authors make hypotheses or claims that are extrapolated too far.
3) The methods section is not as descriptive as it needs to be for it to be repeatable.
a) How were the three individuals sampled for each stage taken from the different families? Did you only use one family in the end?
b) Was there a reason only certain stages of the gonad were chosen for classification?
c) Software for read trimming should be included in the methods. In general, information about the parameters and versions of the used software should be included. Were all reads used from testes and ovaries during the transcritpome assembly (this should be noted)? What software and method was used for annotating the transcripts? How was the BUSCO analysis performed (what database)? What software was used to generate FPKM metrics and identify DEGs. Parameters used for GOseq and KOBAS? Corset should probably be mentioned in the methods rather than the results (with citation).
There are also minor issues that I would like to be addressed before publication:
1) There are grammar issues throughout the manuscript that should be addressed.
2) Some details in the results are not important or relevant to the rest of the study.
3) Figures 4 and 5 needs more detail. It would be good to label the figures so the reader doesn't need to refer to the text. The legend for figure 6 is also not descriptive enough.
4) I don't think the evidence in the results really suggest important single genes, but the discussion is mainly related to single genes being important. If single genes are going to be discussed, the gene's transcription should be characterized in the results.
An optional change:
1) RT-qPCR validation could be as supplementary material.
Round 2
Reviewer 2 Report
While the authors fully addressed my concern regarding the methodology by adding further details, the other two major concerns I had remain (please see previous comments).
In addition, there are still grammar issues that were made even worse with the added tracked changes. I should note that I could not read any of the comments on the side of the PDF as they were cutoff. Again, there were results that really do not add anything to the manuscript.
